# Characterization of the Diagnostic Performance of a Novel COVID-19 PETIA in Comparison to Four Routine N-, S- and RBD-Antigen Based Immunoassays

**DOI:** 10.3390/diagnostics11081332

**Published:** 2021-07-25

**Authors:** Alexander Spaeth, Thomas Masetto, Jessica Brehm, Leoni Wey, Christian Kochem, Martin Brehm, Christoph Peter, Matthias Grimmler

**Affiliations:** 1MVZ Medizinische Labore Dessau Kassel GmbH, Bauhüttenstraße 6, 06847 Dessau-Roßlau, Germany; Alexander.Spaeth@laborpraxis-dessau.de (A.S.); jessica.brehm@laborpraxis-dessau.de (J.B.); Martin.Brehm@laborpraxis-dessau.de (M.B.); 2Institut für Molekulare Medizin I, Heinrich-Heine-Universität Düsseldorf, Universitätsstraße 1, 40225 Düsseldorf, Germany; thomas.masetto@hhu.de (T.M.); christoph.peter@uni-duesseldorf.de (C.P.); 3DiaSys Diagnostic Systems GmbH, Alte Straße 9, 65558 Holzheim, Germany; Leoni.Wey@diasys.de (L.W.); Christian.Kochem@diasys.de (C.K.); 4Hochschule Fresenius, University of Applied Sciences, Limburger Straße 2, 65510 Idstein, Germany

**Keywords:** COVID-19, SARS-CoV-2, quantification of immune response, assay comparison, CLIA, ELISA, PETIA

## Abstract

In 2019, a novel coronavirus emerged in Wuhan in the province of Hubei, China. The severe acute respiratory syndrome coronavirus 2 (SARS-CoV-2) quickly spread across the globe, causing the neoteric COVID-19 pandemic. SARS-CoV-2 is commonly transmitted by droplet infection and aerosols when coughing or sneezing, as well as high-risk exposures to infected individuals by face-to-face contact without protective gear. To date, a broad variety of techniques have emerged to assess and quantify the specific antibody response of a patient towards a SARS-CoV-2 infection. Here, we report the first comprehensive comparison of five different assay systems: Enzyme-Linked Immunosorbent Assay (ELISA), Chemiluminescence Immunoassay (CLIA), Electro-Chemiluminescence Immunoassay (ECLIA), and a new Particle-Enhanced Turbidimetric Immunoassay (PETIA) for SARS-CoV-2. Furthermore, we also evaluated the suitability of N-, S1- and RBD-antigens for quantifying the SARS-CoV-2 specific immune response. Linearity and precision, overall sensitivity and specificity of the assays, stability of samples, and cross-reactivity of general viral responses, as well as common coronaviruses, were assessed. Moreover, the reactivity of all tests to seroconversion and different sample matrices was quantified. All five assays showed good overall agreement, with 76% and 87% similarity for negative and positive samples, respectively. In conclusion, all evaluated methods showed a high consistency of results and suitability for the robust quantification of the SARS-CoV-2-derived immune response.

## 1. Introduction

At the time of writing, more than 184 million people had been infected by SARS-CoV-2 [1,2,3,4] worldwide, and almost 4 million deaths related to COVID-19 had been registered (data from John Hopkins University), resulting in societies and health systems all over the world having to face challenges not known to humanity for many decades [5,6,7]. To manage the pandemic at both the national and international levels, first-line diagnosis, monitoring and care, and vaccination strategies are indispensable [8].

Diagnosis of the SARS-CoV-2 infection [1,2,3] is mainly performed using two different methodical strategies: the direct detection of the pathogen in nasal or pharyngeal swabs, and the indirect detection of the anti-virus antibodies in the blood.

Testing for acute and active virus infections relies on the direct detection of SARS-CoV-2-specific genetic material by reverse transcription polymerase chain reaction (rRT-PCR) [9,10].

To evaluate past infections or the response to vaccination, serological tests on virus-specific antibodies focused on the viral nucleocapsid proteins (N), the spike protein (S), and the receptor-binding domain of the spike protein (RBD) are performed [11,12,13,14,15]. The N-protein is the most abundant protein in SARS-CoV-2. Antibodies to the viral N-protein decline faster than those to the receptor-binding domain or the entire spike protein, and therefore may substantially underestimate the proportion of SARS-CoV-2 exposed individuals [16]. Spike proteins are large membrane-anchored proteins that assemble to form trimers on the surface of the virus (crown-like appearance). Each spike monomer contains a receptor-binding domain (RBD) in the N-terminal S1 subunit, which is responsible for binding to the ACE2 receptor (angiotensin-converting enzyme 2) on the host cell. Interactions between the RBD in subunit S1 and the ACE2 receptor lead to large-scale structural rearrangements of the spike protein, which is essential for virus entry [17,18,19]. Neutralizing antibodies target the highly dynamic S protein, especially on the site of the RBD. Therefore, they are predesignated to induce protective immunity against viral infections [16].

The rapid, inexpensive, and easy-to-scale-up nature of immunoassays has contributed to an increase of anti-SARS-CoV-2 antibody testing frequency. Serological tests help to assess the seroprevalence in a population, or to monitor the exposure of risk groups [20,21], as well as to understand the neutralization potential [22,23] and protective immunity against a viral infection [24,25,26]. They also assist in providing a more accurate picture of the progression of the pandemic in terms of variants, and support the development and monitoring of antiviral drugs and vaccines [16,27].

To date, a broad variety of heterogeneous techniques, such as enzyme-linked immunosorbent assays (ELISA) and chemiluminescent-based type immunoassay (CLIA), have become available to assess and quantify the specific antibody response of a patient toward a SARS-CoV-2 infection. In this article, we present the use of homogenous PETIA (particle-enhanced turbidimetric immunoassay) technology, which has recently been made available for serological quantification. A major aspect of this study was to evaluate whether the PETIA analytical technology has an impact on the diagnostic performance. While ELISA and (E)CLIA are heterogeneous technologies, PETIAs are homogeneous assays. ELISA usually employs a pair of primary antibodies to immobilize the antigen onto the solid phase (capture antibody). Subsequently, the sample is added, and unspecific binding substances are washed away. A second antibody (tracer) carrying the detection molecule is finally added to the reaction. CLIA technology works in a similar way, to a large extent. The major difference from ELISA is the employment of suspended magnetic beads linked to the capture antibody, instead of having it bound to the ELISA well. On the other hand, PETIAs exploit a completely different principle: the specific antibodies are linked to polystyrene beads. After adding the sample, the reaction of the antibodies with the antigen leads to the agglutination of the polystyrene particles, and an increase in the reaction turbidity. The biggest difference between heterogeneous and homogeneous assays is the use of one or more washing steps, which could result in better signal-to-noise ratios and an increase in specificity.

Besides the principle methodical differences, all the assays presented in this work significantly differ in the antigens (N-, S1- or RBD) employed to quantify the SARS-CoV-2-specific immune-response of a patient.

A further major question to address in this context is the sensitivity and specificity of the available assays in different cohorts, such as infants or pregnant women, with their respective altered immunoglobulin backgrounds [28,29]. Moreover, the seroconversion—the specific class switch of antibodies upon the progression of a SARS-CoV-2 infection or vaccination—is a fundamental aspect that we would like to further investigate [12].

There are seven known human pathogenic coronaviruses (HCoV). Four of these species circulate endemically worldwide (HCoV-229E, HCoV-NL63, HCoV-HKU1, and HCoV-OC43). They predominantly cause mild colds but can sometimes cause severe pneumonia in early childhood and elderly individuals [30,31,32]. Due to the broad distribution of the mentioned SARS-CoV-2 relatives, the evaluation and differentiation of “cross-infected” patients are necessary to ensure specific recognition by immuno-tests, such that they do not respond to other *coronaviridae*.

The thorough characterization of five common serological immunoassays presented here considers all raised aspects, especially regarding the impact of assay technology (ELISA, CLIA-ECLIA, PETIA) and the antigens utilized (S1-domain, Nucleocapsid and RBD-domain) in SARS-CoV-2 specific diagnostic performances.

## 2. Materials and Methods

In this retrospective study, leftover samples from 202 SARS-CoV-2-IgG positive and negative patients, plus 20 control samples taken before the SARS-CoV-2 outbreak in Germany (10 pregnant women and 10 children, aged 1–6 years), originating from standard diagnostic laboratory routine procedures, were aliquoted and stored at −80 °C for platform comparison studies. Seroconversion plasma samples obtained from Access Biologicals (Vista, CA, USA) (n = 14) were collected during the progression of one patient’s (male, age: 44) SARS-CoV-2 infection over 87 days. Cross-reactivity studies were conducted with the common corona serum panel from in.vent Diagnostica GmbH (Hennigsdorf, Germany) (Table 1) and patient serum samples with various antibody titers (Table 1, Appendix A). To assess a matrix effect on the test performance, five SARS-CoV-2-IgG negative samples from diagnostic laboratory requests containing serum and coagulated plasma (sodium-citrate, K2-EDTA, and sodium-fluoride, respectively) were tested.

### 2.1. Assays and Instruments

The PETIA method (SARS-CoV-2 UTAB FS, RBD-based antigen) was performed on the c502 Cobas 8000© (Roche Diagnostics, Mannheim, Germany), which required a 4-point calibration with two levels of internal quality controls: level 1 (negative) and level 2 (positive). All materials were liquid-stable and ready to use. Reagents, calibrators, and quality controls were distributed by DiaSys Diagnostic Systems GmbH (Holzheim, Germany). The Anti-SARS-CoV-2-ELISA IgG (S1-antigen-based, EUROIMMUN, Luebeck, Germany) was conducted with full automation, using the Freedom EVOlyzer^®^ (Tecan Group, Maennedorf, Switzerland) according to the manufacturer’s instructions. Three CLIA methods were carried out, which were also fully automated: Liaison^®^ SARS-CoV-2 S1/S2 IgG using the Liaison^®^ XL (S1/S2-based; DiaSorin, Dietzenbach, Germany), Elecsys^®^ Anti-SARS-CoV-2 (N-based) and Elecsys^®^ Anti-SARS-CoV-2-S (RBD-based), both on the e601 Cobas 8000© (Roche Diagnostics, Mannheim, Germany). For a summary of further specifications of the tests used, please refer to Appendix A.

### 2.2. Clinical Performance/Studies

Regarding platform comparison and cross-reactivity studies, one aliquot of each sample was thawed at room temperature and analyzed on all platforms on the same day. The results were evaluated according to the manufacturer’s instructions: for Elecsys^®^ Anti-SARS-CoV-2 (Roche) as not reactive (<1.0 COI) or reactive (≥1.0 COI); for Elecsys^®^ Anti-SARS-CoV-2-S (Roche) as negative (<0.8 U/mL) or positive (≥0.8 U/mL); for Anti-SARS-CoV-2-ELISA IgG (Euroimmun) as negative (<0.8 ratio), intermediate (0.8–1.1 ratio) or positive (≥1.1 ratio); and for Liaison^®^ SARS-CoV-2 S1/S2 IgG (DiaSorin) as negative (<12.0 AU/mL), intermediate (12.0–15.0 AU/mL) or positive (≥15.0 AU/mL).

Intra- and inter-assay precisions of the novel PETIA assay were evaluated by analyzing a positive and intermediate serum, as well as the two levels of quality controls provided by the manufacturer (20 aliquots each).

The hook effect and linearity were assessed using 10 samples with defined increasing SARS-CoV-2-IgG concentrations (0–1562.5 AU/mL). Samples above the highest calibrator concentration (160 AU/mL) were diluted 5- or 10-fold, and the recalculated results were used to accomplish the quantification.

Sample stability was evaluated with three positive SARS-CoV-2-IgG serum samples (78, 85, and 111 AU/mL) which were immediately measured after bleeding, after 7 days and after 14 days, and were stored in aliquots at 4 °C and −20 °C, respectively (Table 2).

### 2.3. Statistical Analysis

Calculation and statistical analyses were performed using XLSTAT^®^ software, version 2016.06.35661 (New York, NY, USA) and MedCalc^®^ Version 18.10.2–64-bit (MedCalc Software Ltd., Ostend, Belgium), the “Diagnostic test” tool, following the principles of C24A3E-Statistical Quality Control for Quantitative Measurement Procedures: Principles and Definitions; Approved Guideline–Third Edition.

## 3. Results

### 3.1. General Test Performance

The novel PETIA assay, which recognizes the RBD for the S-Protein, was compared to four fully automated tests (one ELISA, three CLIA) from different manufacturers. All samples were analyzed on all platforms on the same day for unbiased comparison.

The SARS-CoV-2 UTAB FS assay was performed on the Cobas^®^ 8000 c502-module system (Roche Diagnostics). The first results were obtained 11 min after sample loading and quantified using a four-point calibration curve. Two control levels (positive and negative) were used for daily QC to assure trueness. The QC results were always recovered within their expected reference ranges in both inter-series precision cycles (Figure 1A). Based on these data, the calibration stability was proven for at least one week. Moreover, repeatability and reproducibility studies of a positive patient serum sample (mean 48.59 AU/mL) showed high accuracy, with low CVs in both intra-series (3%) and inter-series (4%) precision. A second intermediate patient serum sample (mean 13.11 AU/mL) showed higher CVs in both intra-series (11%) and inter-series (13%) (Figure 1B,C).

The hook effect and linearity of the PETIA test were analyzed with samples containing increasing antibody concentrations. Within the calibrator range (0–160 AU/mL), we found clear linearity with R^2^ = 0.9989 (Figure 2A). With increasing antibody concentrations above the calibrator range, results tended to develop a plateau around 300 AU/mL.

When diluting these samples 10-fold with the appropriate SARS-CoV-2 UTAB FS dilution matrix, we calculated the actual antibody concentrations. This dilution allowed an extension of the assay linearity beyond the calibration range, up to a concentration of 1562.5 AU/mL with R^2^ = 0.9861 (Figure 2B).

### 3.2. Assay Comparison Study

Patient serum samples were tested simultaneously on four different test platforms besides the novel SARS-CoV-2 UTAB FS PETIA, on the same day. When defining the obtained SARS-CoV-2 UTAB FS values of <10 AU/mL as negative, 10–35 AU/mL as intermediate, and >35 AU/mL as positive, similar qualitative results were found in 77% of all samples (n = 202) over all test platforms, with the highest accordance to both Roche assays (89% each) (Figure 3A). Negative samples showed an overall similarity of 76% (Figure 3B, middle column), but positive samples revealed 87% similarity (Figure 3B, left column). However, the greatest deviations were seen with the intermediate samples (n = 25), resulting in an overall similarity of only 36% (Figure 3B, right column). For further analysis, intermediate values were considered as negative.

The quantitative analysis in terms of the scatter diagram is reported in the Appendix A.

### 3.3. Seroconversion Study

We further investigated the seroconversion of one patient from day 0 to day 87 post-infection, and found antibody levels confidently considered to be positive after 36 days post-infection for SARS-CoV-2 UTAB FS, and 50 days for all the other platforms (Figure 4A–E). Even though the ECLIA technology is, in principle, a more sensitive technique in comparison to PETIA, the supplier does not clearly state the recognition of all isotypes; there is only a vague mention that the assay can also measure IgG. On the other hand, since the SARS-CoV-2 UTAB FS recognizes every existing anti-Spike-RBD immunoglobulin, including potential IgM and IgA, this could encode for the “high-sensitivity” assay behavior (Figure 4A). The maximum level in all assays, except for the Elecsys^®^ Anti-SARS-CoV-2 Nucleocapsid-based assay (Roche, Figure 4E), was reached after 64 days of building a plateau, followed by a slow decrease, respectively.

### 3.4. Influence of Sample Composition and Other Viruses on the Test Performance

Coagulated plasma is not commonly used in clinical chemistry, and is not recommended according to the manufacturers’ instructions. However, to investigate a potential matrix influence of anti-coagulants, five SARS-CoV-2-negative serum samples were compared to their corresponding anti-coagulated plasma samples (sodium-citrate, K2-EDTA, and sodium-fluoride plasma) (Figure 5). We found no signals in the serum samples, as expected, but in contrast, found high signals for all coagulated plasma samples. Even Li-heparinized plasma showed slightly increased signals in some samples, possibly causing false-positive results. Na-citrate coagulant causes the highest signals, closely followed by K2-EDTA plasma and NaF-plasma, with a slightly lower impact.

To reveal any cross-reactions with other antibodies, an infection panel with antibodies against various viral pathogens was tested (Figure 6), as well as sera with antibodies against various unrelated viruses (Appendix A). Despite some panel members revealing a weak basic signal, all tested sera were considered negative, since any value was below the cut-off (<35 AU/mL) of the PETIA assay. Additionally, serum samples from pregnant women and children collected before the SARS-CoV-2 outbreak were tested. While no positive results were seen for the children (n = 10), we found two positive samples from the pregnant women (n = 10) (Appendix A). For these samples, we found no correlation between the positive infection serology and the obtained signals regarding cross-reaction. All sera of pregnant women were additionally quantified regarding the IgM and IgG content of CHPN, MUM, VZV, MAS, CMV, VCA and EBNA to assess the prevalence and status of acute or former infections. The detailed results are reported in the Appendix A.

## 4. Discussion

To date, COVID-19 remains a major threat to global health, and continues to be the main challenge for healthcare systems globally. Fast and accurate diagnosis of a SARS-CoV-2 infection, and monitoring the effectiveness of treatment, places laboratory medicine at the center of patient care, while vaccination strategies will be the long-term solution to the pandemic. Therefore, monitoring of the antibody titer due to a previous infection or upon vaccination will have an essential role in the worldwide management of SARS-CoV-2. However, serology of SARS-CoV-2 has many challenges for laboratory medicine, including the standardization and performance validation of different assays [33,34].

In this evaluation, we characterized the performances of different assay principles for the serological analysis of SARS-CoV-2. The tests of Euroimmun (ELISA), DiaSorin (CLIA), and two Roche (ECLIA) are based on heterogeneous technologies. They use a pair of antibodies to immobilize the antigen onto the solid phase (ELISA wells or magnetic beads), and to detect SARS-CoV-2–specific antibodies of a sample. On the other hand, in the homogeneous PETIA of DiaSys, the specific antigen molecules (RBD) are directly coupled onto polystyrene beads without using a mediating antibody. The RBD antigen directly reacts with the antibodies of the sample. This results in an increase of the reaction turbidity, which can be photometrically measured. The biggest difference from the heterogeneous assays is the lack of washing steps, which might result in lower signal-to-noise ratios and a reduced specificity if PETIA is not properly controlled. In addition, due to direct coupling of the RBD antigen, PETIA detects every antibody isotype (IgM, IgG, IgA), regardless of their affinity for the antigen. It is noteworthy that the IgM’s pentameric structure has a particularly strong effect on the turbidimetric reaction, despite the relatively low affinity of this kind of immunoglobulin. This also explains some of the differences presented in the present work (see Section 3.3 Seroconversion Study).

A further aspect of this study was evaluation of the impact of the SARS-CoV-2´s antigens employed by the manufacturers. The two Roche assays use N and S (RBD) antigens. DiaSorin uses the whole Spike protein (domains S1 and S2). DiaSys exclusively employs the RBD domain. Euroimmun utilizes the S1 domain, containing the RBD. All assays fulfilled general performance criteria (precision, linearity, and the hook effect), and sensitivity and specificity requirements. The overall diagnostic agreement between the five assays was good, at 76% and 87% similarity for negative and positive samples, respectively.

Different cohorts of infants, pregnant women, and a seroconversion panel over 87 days showed similar performances between all assays. No cross-reactivity with common coronaviruses (HCoV-229E, HCoV-NL63, HCoV-HKU1, and HCoV-OC43) occurred. The further evaluation of six unrelated viral pathogens did not lead to unspecific cross-reactions with any tests of this study.

While serum is likely the sample material of choice, all tested heterogeneous assays (CLIA, ECLIA, ELISA) work with plasma as well. However, in exchange for a shortened time to obtain results, the homogenous PETIA lacks washing steps; consequently, plasma samples cannot be recommended for this technology.

PETIA, ECLIA-CLIA, and ELISA-based assays do achieve comparable results when samples are stored at 4 °C or frozen at −20 °C for 7–14 days. Thus, for low requests or re-measurements, archived samples can be measured batchwise, or when requested, without changing the values.

At the time this study was performed, no higher-order reference material was available. Despite the lack of standardization, all evaluated assays reflected the immune status of patients in response to a previous infection properly and comparably. Very recently, WHO (NIBSC 20-136, the first WHO International Standard Anti-SARS-CoV-2 Immunoglobulin (human)) and independent ERM (EURM-017, human serum-antibodies-against-SARS-CoV-2) reference materials have been established for the standardization of immuno-assays.

Another aspect of SARS-CoV-2 specific immunoassays is the ability of the tests to measure neutralization antibodies. At the time of measurement, no neutralization test was available. Human neutralizing antibodies target the host RBD of the SARS-CoV-2 spike protein [35], and can exert their activity by preventing the virion from binding to the ACE2 receptor on the host cells. Binding also causes aggregation of the viral particles, or leads to opsonization and lysis of the viruses [36]. As all the assays here (except Roche N) are based on or contain the RBD domain, their ability to target neutralizing antibodies can be assumed. A systematic correlation of these immunoassays with the virus neutralization test [37,38,39] should be analyzed in future, to more clearly characterize the differences between the test principles.

Everyday use and handling vary greatly among the different evaluated assay types. In particular, the ELISA-principle has a long time to obtain results (150 min) and it needs dedicated instruments.

In contrast, the PETIA principle of DiaSys is flexible and applicable to every analyzer platform. Likewise, the workflow and handling of liquid calibrators, controls, and the on-board stability of PETIA reagents allows for easy and convenient routine application (see Appendix A for more details).

## Figures and Tables

**Figure 1 diagnostics-11-01332-f001:**
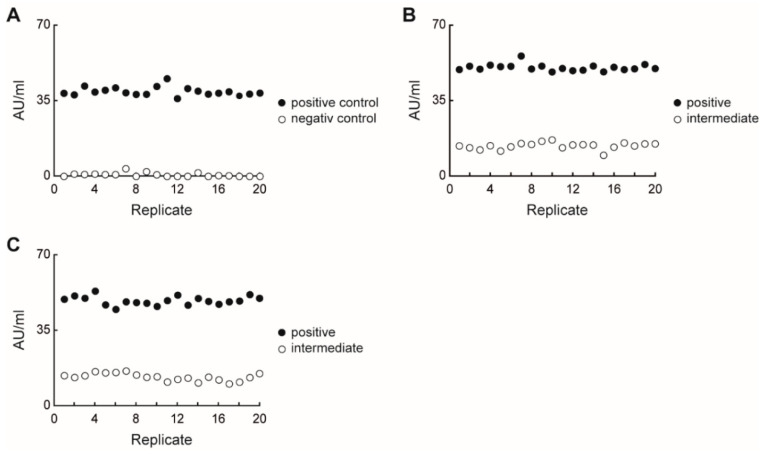
SARS-CoV-2 UTAB FS recovery of quality control (**A**), intra-assay precision (**B**), and inter-assay precision (**C**) with a positive and an intermediate serum sample, showing good accuracy and reagent stability.

**Figure 2 diagnostics-11-01332-f002:**
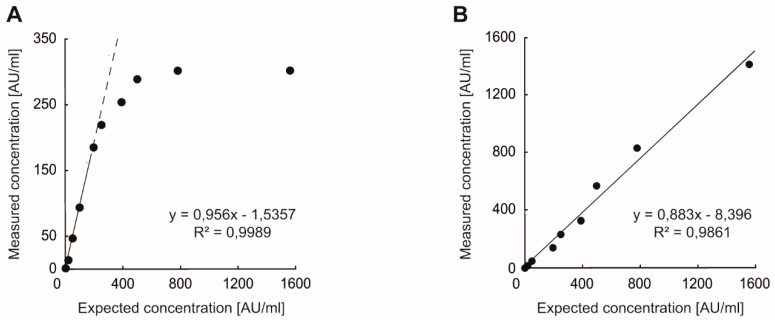
Analysis for hook effect and the high dose–response (**A**) and assay linearity after 10-fold dilution (**B**) revealed assay linearity up to 1562.5 AU/mL.

**Figure 3 diagnostics-11-01332-f003:**
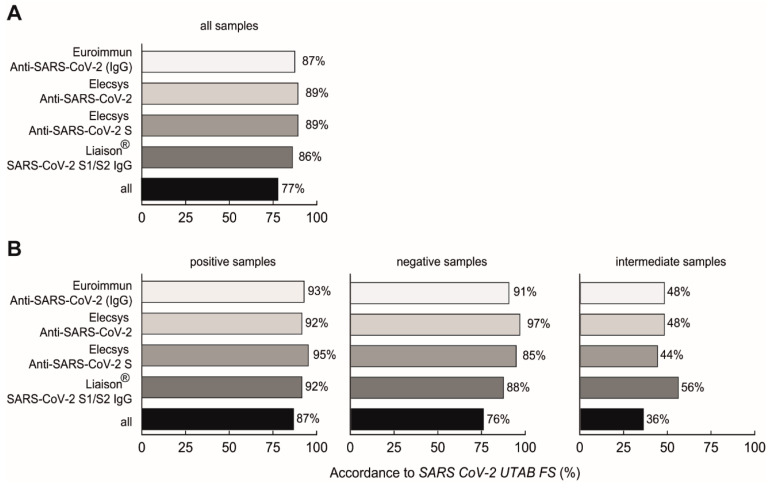
SARS-CoV-2 UTAB FS assay accordance compared to four different SARS-CoV-2 IgG assays for all samples (**A**) and split into positive (>35 AU/mL), negative (<10 AU/mL) and intermediate (10–35 AU/mL) samples (**B**).

**Figure 4 diagnostics-11-01332-f004:**
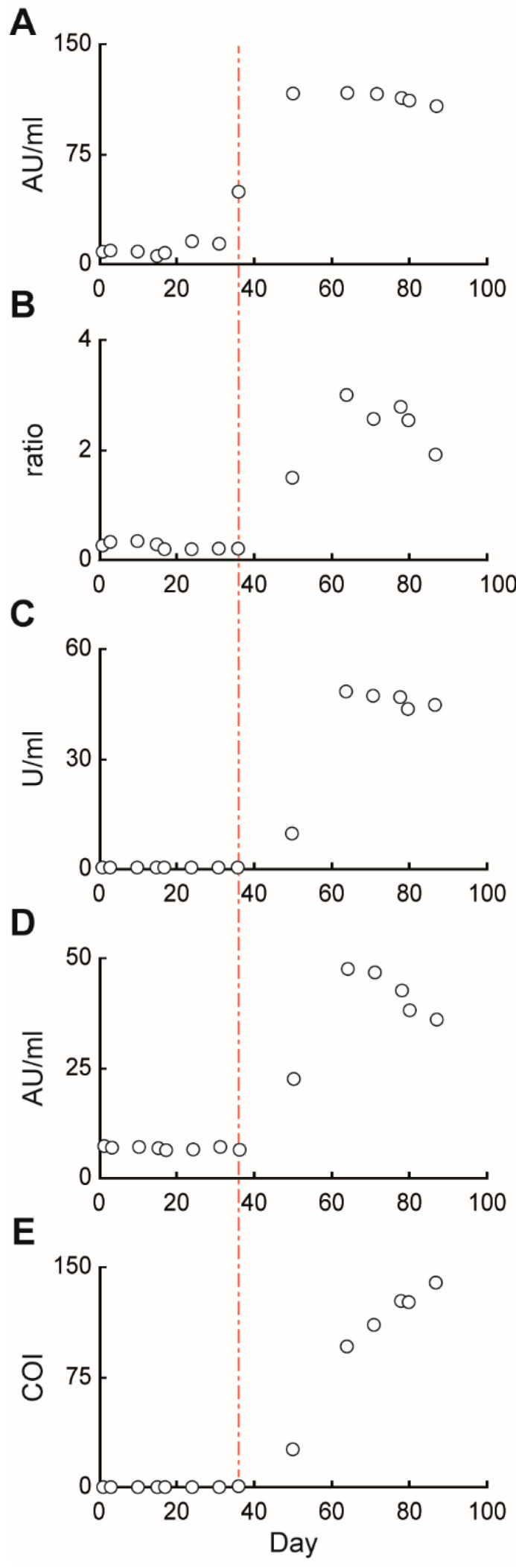
Monitoring of SARS-CoV-2 seroconversion 0 to 87 days post-infection with SARS-CoV-2 UTAB FS (**A**), Anti-SARS-CoV-2-ELISA (IgG) (**B**), Elecsys Anti-SARS-CoV-2 S (**C**), Liaison^®^ SARS-CoV-2 S1/S2 IgG (**D**), and Elecsys Anti-SARS-CoV-2 N (**E**). Seropositivity with the SARS-CoV-2 UTAB FS assay occurred after 36 days (red dashed line), while all other assays became positive after 50 days.

**Figure 5 diagnostics-11-01332-f005:**
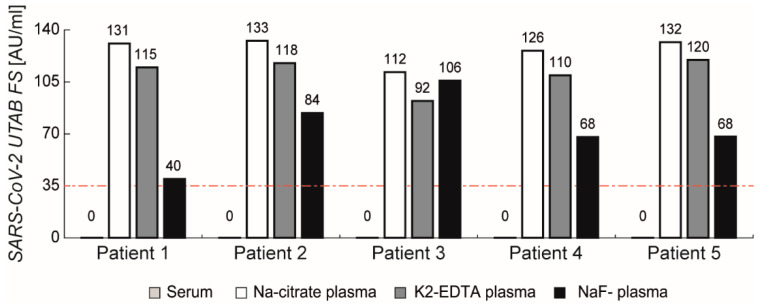
Influence of different plasma matrices on SARS-CoV-2 UTAB FS results compared to serum. All three anticoagulants showed increased signals above the cut-off (red dashed line; 35 AU/mL).

**Figure 6 diagnostics-11-01332-f006:**
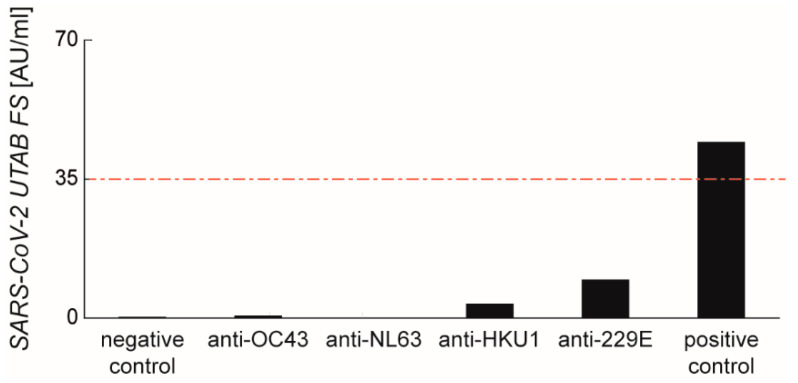
Results of cross-reactivity studies for infection panel samples on SARS-CoV-2 relatives. The four tested human coronaviruses, OC43, NL63, HKU1 and 229E, showed no relevant cross-reactivity with the SARS-CoV-2 UTAB FS assay. The red dashed line marks the cut-off at 35 AU/mL.

**Table 1 diagnostics-11-01332-t001:** Infection panels with selected positive serology used for cross-reactivity studies.

**Kind of Sample**	**Virus**	**Subtype**
**in.vent Diagnostica**		
*Human Donor serum:*	Anti-229E	(alpha coronavirus)
	Anti-HKU1	(beta coronavirus)
	Anit-NL63	(alpha coronavirus)
	Anti-OC43	(beta coronavirus)
**laboratory routine serum samples**	**Positive serology**	
	*Chlamydia pneumoniae*	
	EBV	
	CMV	
	Mumps	
	VZV	
	Measles	

EBV, Epstein–Barr virus; CMV, cytomegalovirus; VZV, varicella-zoster virus.

**Table 2 diagnostics-11-01332-t002:** **:** Sample stability under five selected storage conditions. Three samples (1–3) were measured after 7 and/or 14 days under different storage conditions (A–E). Aliquots for B–E were directly derived after measurement from aliquot A.

			Results (AU/mL)		
Aliquot			0 Days	7 Days	14 Days
		1 ^b^	74.71		
A:	fresh	2 ^c^	87.64		
		3 ^d^	108.41		
		1		80.61	79.34
B:	4 °C	2		93.56	89.8
		3		109.9	108.26
		1			86.87
C:	4 °C	2			77.9
		3			102.04
		1		75.5	75.03
D:	−20 °C ^a^	2		88.07	77.09
		3		118.71	116.5
		1			74.24
E:	−20 °C	2			81.51
		3			112.1

^a^ Aliquots were re-frozen. ^b^ 78.04 ± 4.61 AU/mL. ^c^ 84.97 ± 6.46 AU/mL. ^d^ 110.85 ± 5.57 AU/mL.

## Data Availability

All the data are available upon approval of the corresponding author.

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
