# Peer review of "Characterization of the Diagnostic Performance of a Novel COVID-19 PETIA in Comparison to Four Routine N-, S- and RBD-Antigen Based Immunoassays"

_diagnostics, 2021, doi:10.3390/diagnostics11081332_

Round 1
Reviewer 1 Report
This manuscript presented characters of five assay systems for SARS-CoV-2 serological test. Similar qualitative results were confirmed for all five systems. I found interesting such high consistency.
Main result and conclusion are required in Abstract. It is difficult to read the whole picture from this abstract.
Figure 3 is qualitative comparison. Since 3 assays out of 5 were quantitative, quantitative comparison (scatter plot) is recommended for more informative visualization.
Only SARS-CoV-2 UTAB FS showed positive after 36 days in figure 4. Author's interpretation for the result is that SARS-CoV-2 UTAB FS recognize IgG/IgM/IgA. However, according to Supplement table S1, Elecsys Anti-SARS-CoV-2 S is also recognize IgG/IgM/IgA and its method is ECLIA. In general, ECLIA is the most sensitive assay. Based on these facts, I cannot understand author's interpretation. More interpretation is required.
Please fix "Error! Reference source not found.".
Author Response
- This manuscript presented characters of five assay systems for SARS-CoV-2 serological test. Similar qualitative results were confirmed for all five systems. I found interesting such high consistency. Main result and conclusion are required in Abstract. It is difficult to read the whole picture from this abstract.
We agree with reviewer 1 and we added the main results and a conclusion to the abstract of the manuscript, addressing overall suitability of different methods and used antigens for detection/quantification of SARS-Cov2 specific immune-response.
- Main result and conclusion are required in Abstract. It is difficult to read the whole picture from this abstract.
We integrated main results and conclusions in the Abstract.
- Figure 3 is qualitative comparison. Since 3 assays out of 5 were quantitative, quantitative comparison (scatter plot) is recommended for more informative visualization.
As all methods at the time of evaluation have not been standardized and did use different test-specific antigens and individual unit definitions, we decided not to include scatter plots in the original manuscript to avoid confusion; especially in regard of comparing quantitative with qualitative methods. We integrated and discussed the scatter diagrams and the Spearman´s R and their respective limitation in interpretation in the Supplement materials (new Supplement Figure 1). We also mentioned the new supplemental graph within the manuscript (paragraph 3.2 Assay comparison study).
- Only SARS-CoV-2 UTAB FS showed positive after 36 days in figure 4. Author's interpretation for the result is that SARS-CoV-2 UTAB FS recognize IgG/IgM/IgA. However, according to Supplement table S1, Elecsys Anti-SARS-CoV-2 S is also recognize IgG/IgM/IgA and its method is ECLIA. In general, ECLIA is the most sensitive assay. Based on these facts, I cannot understand author's interpretation. More interpretation is required.
We do agree that Roche´s ECLIA technology in principle is a more sensitive technique in comparison to PETIA. Unfortunately, the manufacturer Roche does not state clearly that the test can measure all the isotypes, but it just vaguely mentions that both its assays can also measure IgG, either qualitatively (N test) or quantitatively (S test). E.g. for the N test: “Elecsys® Anti-SARS-CoV-2 is an immunoassay for the in vitro qualitative detection of antibodies (including IgG) to SARS-CoV-2 in human serum and plasma.” According to this definition of Roche, it is not possible to clearly state a better sensitivity of ECLIA upon PETIA technology without further clear specification of the ability of Roche immunoassays to differentiate between the isotypes.
For this, we think that the PETIA assay may be more dedicated to also detect early IgM and IgA population of antibodies. This may give rise to the observed slightly increased signal at 36 h post-infection, DiaSys vs Roche. We will address and discuss this accordingly in the manuscript.
- Please fix "Error! Reference source not found."
We apologize for the inconvenience caused by the cryptic references. This was caused during up- or download process of the manuscript and incompatibility of software versions. We have changed this in the revised manuscript version accordingly. Furthermore, we will provide an additional not-modifiable PDF-version of the manuscript for the revision process.
Reviewer 2 Report
The MS by Spaeth deals with an interesting topic which is the assessment of three different antigen based immuneassays such as, ELISA, CLIA and PETIA. To my knowledge there is not a one-to-one comparison in terms of the molecular basis of the assays, only references to epitopes (S protein) and other SARS-CoV-2 protein components. Moreover, I find almost no refereces in the introduction part about the S protein structure and dynamics, which is as mentioned by the author one of the targets in serological assays. Enclosed some of the standard in the literature (https://doi.org/10.1039/D0NR03969A, https://doi.org/10.3390/ma13235362) .
I noticed an issue with references starting in page 4 through page 6. This shows a clear lack of care during proof-reading. I suggest to present a new MS, and pay attention to it.
Typos:
1) Line 77: RB-domain--> RBD
Author Response
- The MS by Spaeth deals with an interesting topic which is the assessment of three different antigen based immuneassays such as, ELISA, CLIA and PETIA. To my knowledge there is not a one-to-one comparison in terms of the molecular basis of the assays, only references to epitopes (S protein) and other SARS-CoV-2 protein components. Moreover, I find almost no refereces in the introduction part about the S protein structure and dynamics, which is as mentioned by the author one of the targets in serological assays. Enclosed some of the standard in the literature (https://doi.org/10.1039/D0NR03969A, https://doi.org/10.3390/ma13235362).
We agree with Reviewer 2 that the molecular/structural composition of SARS-CoV-2´s proteins is an important topic to substantiate the performance and the comparison of the different evaluated tests. In this direction, we already summarized and compared assay specifications and antigens within the supplement section (Supplement Table 1). Additionally, we have now deepened this topic further in the introduction reporting new data and literature on the structural aspects of the individual assays. Moreover, we included a more focussed part on structural aspects and dynamics of the SARS-CoV-2´s proteins (Spike and Nucleocapsid), picking up different references as suggested by the Reviewer as well.
Furthermore, we introduced a short but comprehensive description of the molecular mechanisms at the base of the ELISA, CLIA and PETIA technologies within the introduction part of the manuscript.
- I noticed an issue with references starting in page 4 through page 6. This shows a clear lack of care during proof-reading. I suggest to present a new MS, and pay attention to it.
We apologize for the inconvenience caused by the cryptic references. This was caused during up- or download process of the manuscript and incompatibility of software versions. We have changed this in the revised manuscript version accordingly. Furthermore, we will provide an additional not-modifiable PDF-version of the manuscript for the revision process.
- Typos: 1) Line 77: RB-domain--> RBD.
We have changed the typo of the sentence accordingly.
- Moderate English changes required.
The manuscript has been cross-checked by native speaker once more to eliminate further typos (related changes are also highlighted within the manuscript itself).
Round 2
Reviewer 2 Report
I am glad the authors have implemented promptly my comments. I endorse the article for publication and wish many successes in the further development of this technology.
Author Response
Thank you very much.